# Influence of Traditional vs Alternative Dietary Carbohydrates Sources on the Large Intestinal Microbiota in Post-Weaning Piglets

**DOI:** 10.3390/ani9080516

**Published:** 2019-08-01

**Authors:** Marco Tretola, Alice Luciano, Matteo Ottoboni, Antonella Baldi, Luciano Pinotti

**Affiliations:** Department of Health, Animal Science and Food Safety, VESPA, Università degli Studi di Milano, 20133 Milano, Italy

**Keywords:** former food products, weaning piglets, gut microbiota, next generation sequencing, gut health, alternative feed ingredients, sustainability

## Abstract

**Simple Summary:**

Nutritional and environmental changes result in significant physiological changes in pigs at the weaning stage. The post-weaning period is mainly characterized by low feed intake and feed efficiency, together with intestinal disturbances. Maximizing the energy intake is known to be critical for promoting growth in weaned piglets, and it is essential to formulate diets with highly digestible and absorbable nutrients/ingredients, as the degree of intestinal maturation is limited. The current challenge is to find new sustainable, effective, and simple carbohydrate sources to satisfy these conditions without producing detrimental effects on the gut ecosystem. In this research, processed and ready-to-eat food products that are no longer suitable for humans were tested, which have high potential as an alternative energy source for pig nutrition. The results demonstrated that replacing conventional ingredients with highly digestible and simple carbohydrate-rich ingredients in the diets of post-weaning piglets did not affect their growth. However, both the abundance and composition of the bacterial community in the large intestine changed. Thus, the results should be interpreted with caution, as they are case-specific, and when these alternative feed ingredients are used in the post-weaning period, their inclusion rate and their effect on microbiota must be carefully considered.

**Abstract:**

In this study, common cereal grains were partially replaced by former foodstuffs products (FFPs) in post-weaning piglets’ diets, to investigate how these alternative ingredients influence the faecal microbiota in the post-weaning period. Twelve post-weaning piglets were housed for 16 days in individual pens and were then fed two diets: a standard wheat-barley-corn meal diet and a diet containing 30% FFPs, thus partially substituting conventional cereals. The growth performance was monitored and faecal microbiota was characterized by the next generation sequencing of the 16S rRNA gene. The results showed no detrimental effects on growth performance when FFPs were used. However, the FFP diet decreased the bacterial richness and evenness in the large intestine, while minor differences were observed in the taxa composition. The core microbiota composition was only slightly affected, and no differences between the two groups in the gut microbiota composition at the phylum level over time were observed. Thus, although these results should be interpreted with caution, as they are case-specific, FFPs can be potentially used as alternative carbohydrate sources in post-weaning piglets, but further investigations are necessary to clarify their impact on gut health when used for a longer period.

## 1. Introduction

The increasing need to find alternative protein/energy sources has triggered research in the field of non-conventional feed ingredients, with former foodstuffs being among the most promising. Former foodstuff products (FFPs) represent a “nutritious” biomass with great potential for farm animals [1]. They are foodstuffs that become unsuitable for human consumption for various reasons, such as production errors leading to broken or intermediate foodstuffs, surpluses that are caused by the logistical challenge of daily deliveries, and surpluses that are caused by the discontinuation of a food product line [2,3,4]. The nutritional and functional characteristics of several categories of FFPs have been recently investigated [4,5,6], and the general conclusion was that these materials could be classified as a “fortified version of common cereal grains”. These aspects have been extensively addressed by Giromini and co-workers [4], who reported that FFPs have a similar nutritional composition to wheat grain, although with a higher energy (metabolisable energy, ME) content. The ME value that was reported for FFPs was 16.95 MJ kg^−1^, where fats and starch represented the main contributor to this energy content. Has been also observed that the ether extract (EE) content of FFPs can be up to six times higher than that of the wheat, and the starch content is also very high, reaching about 52% on a dry matter (DM) basis. Another appealing characteristic of FFPs is its digestibility, which ranged from 79% up to 93% on DM basis in the abovementioned study [4], depending on the ingredients used for their production. However, FFPs, as expected, cannot be considered as a valuable protein source due to their low protein quality and content.

Other authors [5] evaluated nutritional composition of FFPs derived from the bakery industry. Liu et al. [5] found that the average concentration (DM basis) of crude protein (CP, 12.20 ± 2.16%), ether extract (EE, 9.38 ± 1.95%), starch (44.61 ± 5.47%), and neutral detergent fibre (NDF, 13.77 ± 4.23%), indicated that bakery meal consists of a mixture of food ingredients originating from flour or whole cereal grains including some high-fiber ingredients such as brans or canola coproducts. Again, a relatively high fat content was observed, which indicated that oils or fats were added during food production. A further investigation into these materials, which also addressed the kinetics of carbohydrate digestion, was proposed by Ottoboni et al. [6]. They evaluated the predicted glycemic index (pGI) in FFPs, and in two pig compound feeds formulated with or without FFPs. The results clearly indicated that the FFPs and the diets formulated with FFPs up to 30% were characterized by a high glycemic index potential, which appears to be linked to the starch/sugars hydrolysis index. However, this aspect would seem to depend on the treatment received by FFPs. Indeed, as already demonstrated [4], due to the origin of the raw materials (food leftovers) used in the production, FFPs are composed of previously heat-treated or cooked starch. Thus, they are characterized by a higher level of digestibility compared to the cereal grains commonly used in pig nutrition. 

It is known that starch processing can modulate the kinetics of starch digestion [6] and the glycaemic index [7], although the contribution of the simple sugars cannot be ruled out. When compared to common cereal grains, FFPs have higher content of simple sugars (more than 200 g·kg^−1^). These sugars have been linked to a higher digestibility potential measured in vitro, which probably affected the results that were obtained by Ottoboni et al. [6]. Guo et al., [8] recently investigated the effects of a supplemental candy coproduct as an alternative carbohydrate source to dietary lactose, on growth performance, and on the health status of nursery pigs. They found no adverse effects on growth performance or health status when candy coproducts were used to replace up to 18% of conventional feed ingredients in nursery pigs. However, in this case health status was monitored only in terms of mortality and morbidity, blood urea nitrogen, and fecal score. Prandini et al., [9] evaluated the inclusion, up to 80%, of other types of FFP (i.e., dry pasta) in finishing pig diets on growth performance, carcass characteristics, and ham quality. They found that the inclusion of pasta-based FFPs increased the carcass weight, and the dressing percentage reached the highest values at 30% inclusion level. The characteristics of the carcass and the quality of ham were also adequate. When compared with the standard diet, the diets that were based on pasta had a higher predicted glycemic index (+28%), and lower percentages of slowly digestible starch (−65.94%) and resistant starch (−60.73%). These features make these materials suitable for the early nutrition of farm animals, for which digestive physiology, feed formulation, ingredient selection, feed manufacturing, and treatments are essential. In young animals, such as piglets, adapting to dry food is the third phase in the evolution and maturation of gut microbiota, and it starts from weaning [10]. Modern pig production involves this important step very early in the piglet’s life, usually at three or four weeks of life. Consequently, during the first week post-weaning, the microbiota become highly unstable with a strong decrease in bio-diversity, which is then restored after two or three weeks [11]. To the best of our knowledge, a functional evaluation of FFPs that focused on their impact on gut health has never been described or its potential for an efficient animal production. Therefore, the objective of this study was to evaluate the effects of readily fermentable substrates, such as FFPs in the gut microbial community and biodiversity of post-weaning piglets.

## 2. Materials and Methods 

### 2.1. Former Food Product Ingredients

In this study, the FFP diet was formulated to contain a standard FFP product that was provided by an FFP-processing plant based in the north of Italy. Details regarding the ingredients used by the processing plant to obtain the complementary FFP product (namely FFP2), together with its chemical composition, are reported in Giromini et al. [4]. Briefly, it was composed of leftovers from the food industry (bakery products, pasta, of pastry products industry, confectionery products), wheat by-products (e.g., bran), and wheat flour. The nutrient composition on DM basis was: Moisture 8.7%, CP 11.5%, EE 11%, crude fibre (CF) 4.4%, NDF 15.7%, acid detergent fibre (ADF) 5%, Starch 42%, Ash 3.2%, nitrogen free extracts (NFE) 61%, and non-structural carbohydrates (NSC) 59%.

### 2.2. Animals, Housing and Treatment

The Experimental Animal Research and Application Centre in Lodi, at the University of Milan hosted the in-vivo trial, in accordance with the Italian ethical regulation (DL 26/2014, protocol 711/-PR). In addition, the principles of the 3Rs (Replacement, Reduction and Refinement) were applied to the trial authorized by the Italian Health Ministry. Large White × Landrace pigs (n = 12), weighing 8.52 ± 1.73 kg at 28 days were used for this study. Before the start of the experiment, the absence of hemolytic *E. coli* strains in faeces was verified. After one week of adaptation, the piglets were housed for 16 days in individual pens. According to a European Directive (EC Directive 2008/120/EC), environmental enrichment was provided in the form a wooden log on the floor. The piglets were randomly grouped (CTR and FFPs) to obtain similar conditions of initial body weight. Six pigs belonging to the CTR group were fed a standard diet for post-weaning piglets, while for the pigs belonging to the FFPs group (n = 6), the FFPs-based diet was used (Appendix A). Feed and water were provided *ad libitum*. Feed leftover was weighed daily to estimate the average daily feed intake (ADFI, kg/day), while the piglets were weighed on days 0, 5, 9, and 16 to estimate the bodyweight (BW, kg), which was used to calculate the average daily gain (ADG, kg/day) and the feed conversion ratio (FCR, kg/kg). Specifically, ADFI, ADG, and FCR were calculated as the means of the entire experimental period (16 days).

### 2.3. Experimental Diets

The experimental feed ingredients are reported in Appendix A. Briefly, when compared to the standard diet (CTR), in the FFPs-based diet, 30% FFPs were used to partially replace conventional cereal grains (wheat, barley, and corn), together with plasma and whey powder. The chemical composition of the diets was analysed according to the Association of Official Analytical Chemists (AOAC) [12] and the European Commission Regulation No. 152/2009. In addition, glucose, fructose and sucrose were quantified according to the protocol PT 119 NA-2017. Finally, the equation formulated by Noblet and Perez [13] and further adapted by NRC [14] was used to estimate the metabolisable energy (ME) values for pigs.

The energy and chemical constituents that were used in the calculation were expressed on a DM basis in all equations.

The composition of the control (CTR) and FFP based diets is reported in Appendix A. The diets were iso-energetic (16.0 MJ/kg DM) and iso-nitrogenous (20.5% DM), and they met NRC (2012) requirements (Appendix A).

### 2.4. Sample Collection

The faecal samples were collected from rectal ampulla, immediately frozen in liquid nitrogen, and then stored at −80°C until further analysis. Specifically, the samples were collected on days D 0, D 8, D 16 after a one-week adaptation period. In total, 36 faecal samples were sent for Illumina sequencing: six samples per experimental group x two experimental groups x three sampling times (36 samples).

### 2.5. DNA Extraction and Sequencing

The QIAamp Fast DNA Stool Mini Kit (QIAGEN, Germantown, USA) was used to extract bacterial DNA from stool samples, starting with 200 µg of stool following the manufacturers’ procedure. The extracted DNA was quantified using Nanodrop ND2000 with a final concentration ranging from 3–10 ng/uL. Variable regions V3 and V4 of the 16S rRNA were amplified by PCR with universal primers for prokaryotic (341F/802R: CCTACGGGNGGCWGCAG/GACTACHVGGGTATCTAATCC, respectively). The DNA quality assessment and the next generation sequencing (NGS) of the extracted amplicons were both performed by BMR Genomics (Pavia, Italy) on an Illumina MiSeq 300 PE platform to obtain raw paired-end reads 2 × 300 bp.

### 2.6. Growth Performance Data

The data were analysed using IBM SPSS Statistics version 24 (SPSS/PC Statistics 24 SPSS Inc., Chicago, USA). For the performance data, the pig was considered to be the experimental unit. Data were tested for normality with the Shapiro–Wilk test before statistical analysis. Growth performance data (BW, ADFI, ADG, and FCR) were analysed using one-way analysis of variance (ANOVA) in order to compare means. The REPEATED statement was used for variables measured over time (BW, ADFI, ADG, and FCR). Differences with *p* values < 0.05 were considered to be significant.

### 2.7. NGS Data Analysis

The open source pipeline QIIME version 1.9.1 [15,16] was used for the 16S rRNA gene sequences, quality control, and operational taxonomic unit (OTU) binning. The sequences were binned into OTUs based on 97% identity against the Greengenes reference database v. 13–8 [17]. Before the downstream analyses, sequences were rarefied to obtain an equal number of reads for all of the samples in the different time points. The taxonomy at each level was investigated while using the specific “summarize taxa” function in QIIME. The same software was used to analyze the alpha and beta (unweighted and weighted UniFrac metrics) diversities. Analysis of similarity (ANOSIM) was used to evaluate significant differences (*p* < 0.05), together with the effect size of the test (R) in the gut microbiota among diets. Differences in the taxa composition between the two groups of pigs at each time point were investigated through the linear discriminant analysis of the effect size (LEfSe) algorithm while using the Galaxy online tool [18].

In addition, LEfSe uses linear discriminant analysis to estimate the effect size of each differentially abundant feature. The “microbiome” library in R, version 2.5.0 (Boston, USA), was used to estimate the core microbiota, which is the number of common microbes within an arbitrary set of samples. A two-way repeated measures ANCOVA was used to compare the proportions of taxa as well as diversity indexes (Chao1, OTUs, PD-whole tree and Shannon indexes) between the two groups of pigs during the different sampling time points while using a GLM procedure in SPSS (SPSS/PC Statistics 24 SPSS Inc., Chicago, USA, 207 IL).

## 3. Results

### 3.1. Growth Performance

The results of the present trial regarding pigs’ performance were published by Tretola et al. [19]. Briefly, at the end of the experiment, no differences in BW were observed between groups (*p* = 0.61; Figure 1).

Average daily gain (kg) and average daily feed intake (kg) were not affected by dietary treatment (Figure 2). Conversely, piglets on the FFP diet showed a lower feed conversion ratio (*p* < 0.01) compared with control ones. These figures were accompanied by a higher (+4.75%) apparent total tract digestibility recorded for FFP diet compared to the conventional diet.

### 3.2. Gut Microbiota

A total of 506,725, 491,093, and 557,399 sequences were acquired for days 0 (D 0), 8 (D 8), and 16 (D 16), respectively. Details regarding the number of sequences that were obtained for each sample after the quality check of the reads are reported in the Appendix A. Rarefaction curves of each sample in the different time points are shown in Appendix A. The alpha diversity analysis showed that the time did not affect either the number of OTUs or the diversity within a group. However, from D8 onwards, the FFPs decreased (*p* < 0.05) Shannon’s index, together with a decrease (*p* < 0.05) in the OTU average by D 16. Table 1 reports a summary of several alpha diversity indexes of all experimental periods.

When considering the diversity between samples (beta-diversity), the UniFrac distance metric incorporates information regarding the phylogenetic distances between observed organisms in the 16S rRNA dataset. In this study, this metric showed that the animal microbiota was uniform in D 0 (*p* > 0.05), while the use of FFPs led to a qualitative modification in the gut microbial community over time. Specifically, while no changes were observed in D 8, the unweighted UniFrac beta-diversity analysis showed slight clusterization (*p* < 0.05, R = 0.2) in the microbial community between the two dietary groups (Figure 3) in D 16.

While the inter-individual Bray-Curtis distances between different individuals was stable after D8 in CTR group, it significantly changed from D8 to D16 (*p* < 0.01) in FFPs group (Figure 4A,B). 

Regarding the gut microbiota composition in D 16, the LefSe analysis showed that the enriched phylotypes from the CTR group predominantly belonged to the class Bacilli and genus Lactobacillus, whereas phylotypes from the FFP group belonged to the phylum Proteobacteria (Figure 5A,B).

After 16 days of the experiment, the core microbiota—defined as operational taxonomic units (OTUs) found in all samples—were comprised of 66 and 69 OTUs for the CTR and FFP groups, respectively. Figure 6 illustrates the main core microbiota OTUs in CTR and FFPs groups at day 16. 

However, repeated measurement ANCOVA analysis showed no differences between the two groups over time in the gut microbiota composition at the phylum level when the temporal changes between CTR and FFPs are taken into account. Figure 7 shows an overview of the temporal changes in the bacterial community at the phylum level. According to the literature [20], the proportion of Firmicutes significantly increased (*p* < 0.01) between D 8 and D 16 (45.9 ± 2.9% and 65.4 ± 3.5%, respectively) in the pigs fed a standard diet.

Similarly, the main changes in the FFP group over time affected the Bacteroidetes, which increased (*p* = 0.007) during the first period (27.7 ± 2.56% and 48.2 ± 3.49% in D 0 and D 8, respectively), and decreased again to the original values (*p* = 0.008) (29.7 ± 6.01%) in the last sampling time D 16. Figure 8 shows the time-trend of Firmicutes and Bacteroidetes in the two groups.

## 4. Discussion

The use of alternative feed ingredients to replace conventional cereal grains needs to be addressed, not only in term of national value, but also in term of its safety [3]. To our knowledge, this study is the first to address the effects of former foodstuff products on gut microbiota when used to replace 30% of traditional cereal grains in the feed formulation for post-weaning piglets. The results demonstrated that this substitution has no detrimental effects on growth performance [19]. However, in the present study, both the bacterial abundance and its biodiversity, as indicated by the OTU number and Shannon’s index, respectively, decreased in the post-weaning piglets that were fed FFPs, as compared to the CTR group. These results indicate that FFPs impaired the colonization of gut microbiota during the post-weaning period, as highlighted by the alpha diversity analysis. We speculate that results are related to the nature of FFPs. The experimental diets used in this trial have been evaluated for both in vivo and in vitro digestibility in a parallel study [19], where FFP diet showed higher values of both in vitro and in vivo digestibility as compared to the conventional diet. Here, it is important to note that FFPs are obtained by food industry leftover [4,19], starchy-food originally intended for human consumption. Therefore, it is usually subjected to mechanical and/or thermal processing, which can strongly affect starch digestibility [4]. However, starches that are commonly used in the livestock sector are fed largely in a raw form [21]. In addition, our results showed that the different nature of FFPs when compared to untreated cereals used in feed formulation could result in a reduced TDF. It is well known that TDF is the main non-digestible component of monogastric diets and its administration alters the gut environment by providing substrates for microbial growth [22]. Insoluble DF, such as cellulose and hemi-cellulose, have a fecal bulking effect, but they are not or are only marginally digested by the gut microbial population. Soluble fibers are not involved in the fecal bulking, but the gut bacteria ferments them, leading to the production of SCFAs [22].

Accordingly, in our study, a smaller amount of TDF reached the large intestine in piglets that were fed FFPs as compared to those fed standard diet, resulting in a reduced gut bacterial abundance. Alpha diversity showed that not only the bacterial abundance has been affected by the FFPs diet, but also its biodiversity, as indicated by the lower value of the Shannon index, when compared to the CTR group. In fact, the biodiversity of the faecal bacterial community of FFPs and CTR pigs differed as early as day 8, with a decrease in the relative proportion of dominant bacterial species in FFP pigs. A decrease in microbiota abundance and diversity is often related to a decreased ability of bacterial ecosystem to respond to gastrointestinal perturbations [23] and to an increased probability of pathogen colonization in the gut [24]. Stecher et al., [25] showed that both an ecosystem with a low diversity and the abundance of commensal bacteria could reduce the ability of the gut ecosystem to resist to pathogens. On the other hand, Werner et al., [26] demonstrated that the reduction in biodiversity might be associated with ecosystem instability, in the case of abiotic aggression. Thus, we found a less stable faecal microbiota in piglets that were fed FFPs compared to CTR, as indicated by the beta-diversity analysis. The microbial community structure, in fact, changed between the two groups in D 16. In addition, the inter-individual Bray-Curtis distances between different individuals significantly increased during the last period (from D 8 to D 16) in piglets that were fed FFPs diet, while it was stable in CTR during the same period. This result suggested that the gut microbiota structure of piglets fed FFPs became more dissimilar and less stable during the last period, when compared to the CTR group. These differences could be explained by a different grade of maturation in the gut microbiota structure between the two groups. An immature gut microbiota is sensitive to environmental factors and more vulnerable to be disturbed [27]. The speed of maturation of gut microbiota can be variable among each individual, especially around the weaning period, where the gut microbiota structure of piglets became increasingly dissimilar. However, generally, the succession of piglet gut microbiota continues until the establishment of a climax community [27], composed of microbes in stable association with the host and in a stable composition. Piglet gut microbiota may reach a climax community 10 days post-weaning [28]. However, the low bacterial abundance, together with the higher inter-individual Bray-Curtis distances that were observed in our study, could indicate an immature gut microbiota in piglets fed FFPs when compared with piglets that were fed standard diet. Consistent with previous studies, Firmicutes and Bacteroidetes were the two most dominant phyla in piglet gut microbiota, independently by the time point [29,30,31]. After 16 days of the experiment, LefSe analysis highlighted an increased number of bacteria belonging to Proteobacteria phylum and a decreased number of bacteria belonging to the genus *Lactobacillus* in the FFP group, compared to the CTR group. *Proteobacteria* is the most diverse bacterial phylum in the fecal microbiota and its members include several well-known opportunistic pathogens, such as Escherichia coli, Salmonella, and Campylobacter, which may affect the health of the host. Furthermore, in other host species, high abundances of Proteobacteria have been associated with dysbiosis in the hosts with metabolic or inflammatory disorders [32,33,34]. The abundance of the genus *Lactobacillus*, a member of Firmicutes is of particular interest. Our results are in contrast with the literature, which showed a positive correlation between soluble DF and health-promoting bacteria, such as Bifidobacterium, *Lactobacillus,* and Eubacterium [22]. In the present study, despite a higher soluble DF fraction in FFPs diet, the abundance of *Lactobacillus* genus decreased in pigs that were fed FFPs as compared to the standard diet. Probably, this soluble fraction was too readily fermentable to reach the large intestine, with no effect on colonic bacterial population. However, bacteria belonging to the genus *Lactobacillus spp*. are considered as beneficial bacteria for the gut function and health of the host [35] and they have been reported as the most prominent probiotics from the lactic acid bacteria group [36]. Consequently, an increased amount of slowly fermentable DF should be recommended, which can also allow carbohydrate fermentation in the large intestine. In light of these considerations, FFP has also been addressed in different diets/and formulations to explore the existence of positive associative effects with selected dietary fibre sources (e.g., pectins), which can enhance feed efficiency. We characterized a phylogenetic core of gut microbiota, which was made up of a small number phylotypes, despite of the gut microbiota structural variations across the two groups. In this respect, pigs that were fed the CTR and FFP diets showed a similar core microbiota, being composed of a similar number of OTUs and both groups showed Phascolarctobacterium as the most representative OTU of the core microbiota. Phascolarctobacterium belongs to the phylum of Firmicutes and produces short-chain fatty acids, including acetate and propionate. It is commonly found in the human gut, providing beneficial effects to the host [37]. Other key taxa that were observed in both CTR and FFPs are the family S24-7 belonging to the order Bacteroidales, which is a common component of mammals’ gut microbiota [38]; the genus *Roseburia*, belonging to the Lachnospiraceae family, which is part of the commensal bacteria producing short-chain fatty acids, with positive effects on colonic motility, immunity maintenance, and anti-inflammatory properties [39]; and, the genus *Prevotella*, a fibre-utilizing bacteria belonging to Bacteroidetes and associated with high fibre diets. Of the OTUs characterizing the core microbiota of FFP pigs, the genus *Lactobacillus* is absent. In our study, we found differences in the abundance and biodiversity in faecal gut microbiota of piglets that were fed FFPs and standard diet for 16 days, together with slight differences in microbial communities, but no differences on the growth performance have been found.

## 5. Conclusions

By combining the results that were obtained in the present study, it can be concluded that the use of a large volume of FFPs (up to 30%) in complete piglet diets decreased the bacteria abundance, biodiversity, and stability in the large intestine, as indicated by alpha and beta diversity results. Moreover, their use in post-weaning diets affected the relative abundance of both Proteobacteria phylum and the *Lactobacillus* genus, as compared to piglets that were fed standard diet. Concluding, although these results should be interpreted with caution, since they are case sensitive, it can be suggested that FFPs can be used as alternative carbohydrate sources in post-weaning piglets without any detrimental effect on pig growth performance, even if further investigations are necessary for clarifying their impact on gut health and the microbiota ecosystem. Monitoring the effects of the lack of *Lactobacillus* in the core gut microbiota of piglets that were fed FFPs over a longer period would be of benefit, and in other rearing phases, such as growing and finishing.

## Figures and Tables

**Figure 1 animals-09-00516-f001:**
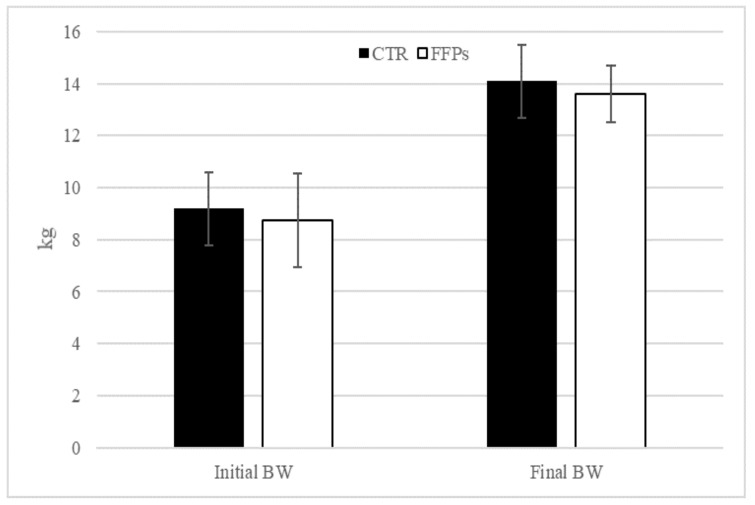
Effects of partial replacement of conventional cereals by FFPs on initial and final body weight. CTR = standard diet; FFPs = former foodstuffs products diet; BW = body weight; FCR = feed conversion ratio; Value for each item is the mean ± SEM (standard error of the mean).

**Figure 2 animals-09-00516-f002:**
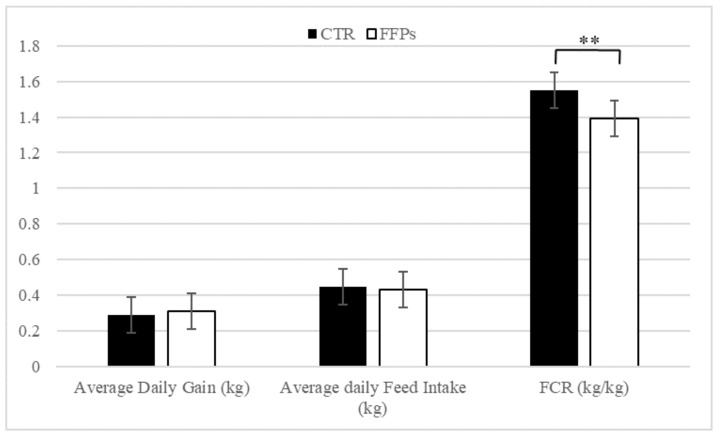
Effects of partial replacement of conventional cereals by FFPs on growth performances. CTR = standard diet; FFPs = former foodstuffs products diet; FCR = feed conversion ratio; Value for each item is the mean ± SEM (standard error of the mean); ** Values differ significantly at *p* < 0.01.

**Figure 3 animals-09-00516-f003:**
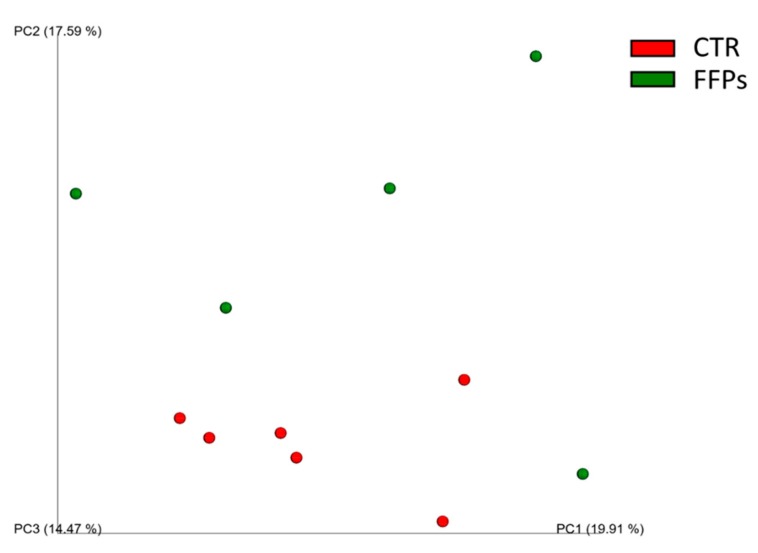
Principal Component Analysis (PCoA) of unweighted UniFrac beta-diversity of gut microbial communities from stools collected at the end of the experimental period. The first three principal coordinates (PC) from the PCoA are plotted. Symbols represent data from individual piglets, color-coded by the metadata indicated.

**Figure 4 animals-09-00516-f004:**
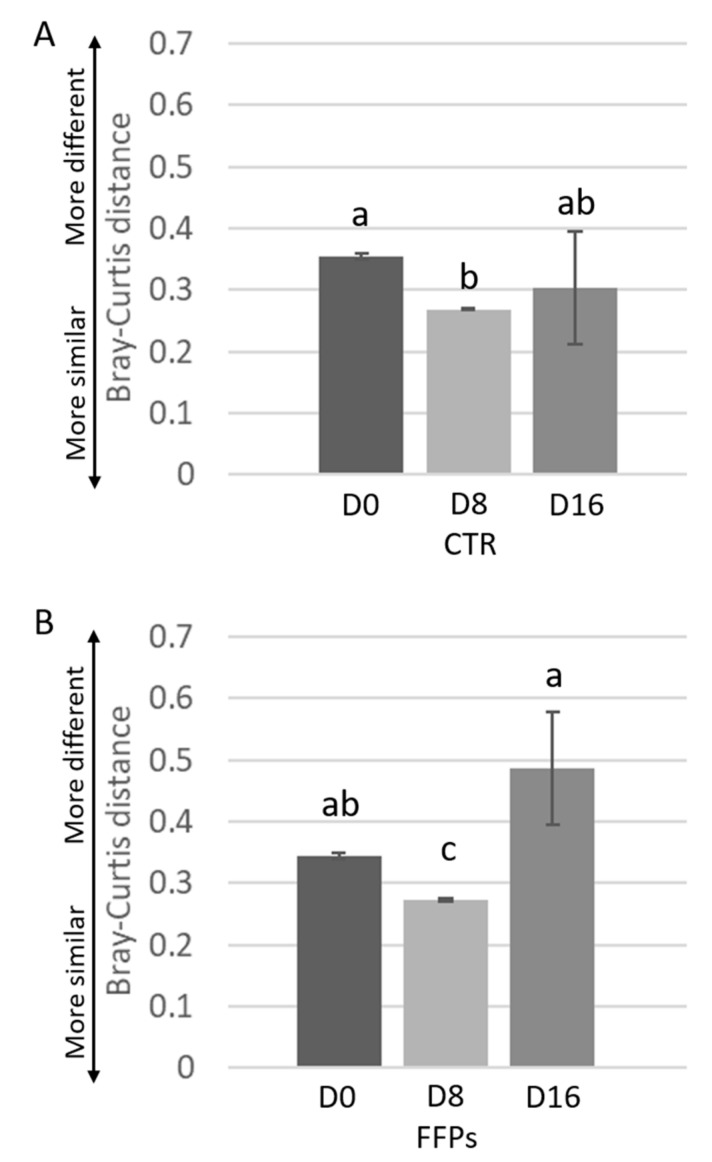
Inter-individual variations of the gut microbiota of the piglets during time. Inter-individual variations were determined by average Bray-Curtis distances between individuals at the beginning of the trial (D 0), after eight days (D 8), and after 16 days (D 16) the use of the experimental diets. Results are shown for pigs fed CTR (A) and FFPs (B) diets. Mean values ± SEM are shown. Different letters above the bar denote significant difference tested by Student’s *t*-test with 1000 Monte Carlo permutations.

**Figure 5 animals-09-00516-f005:**
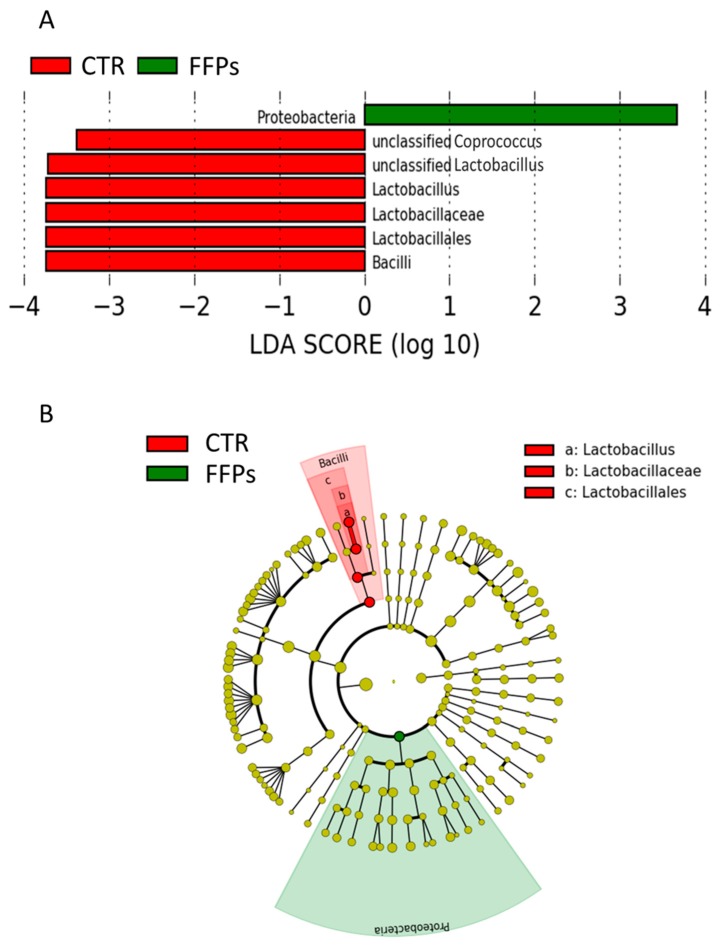
(**A**) Plot from LefSe analysis indicating enriched bacterial taxa associated with post weaning piglets fed with the FFP (green) or CTR (red) diet; (**B**) Different representation of LefSe analysis in the form of cladogram, which is one way of representing significance and phylogeny. The red and green colors represent the branch of the phylogenetic tree that most significantly represents the CTR and FFP groups, respectively.

**Figure 6 animals-09-00516-f006:**
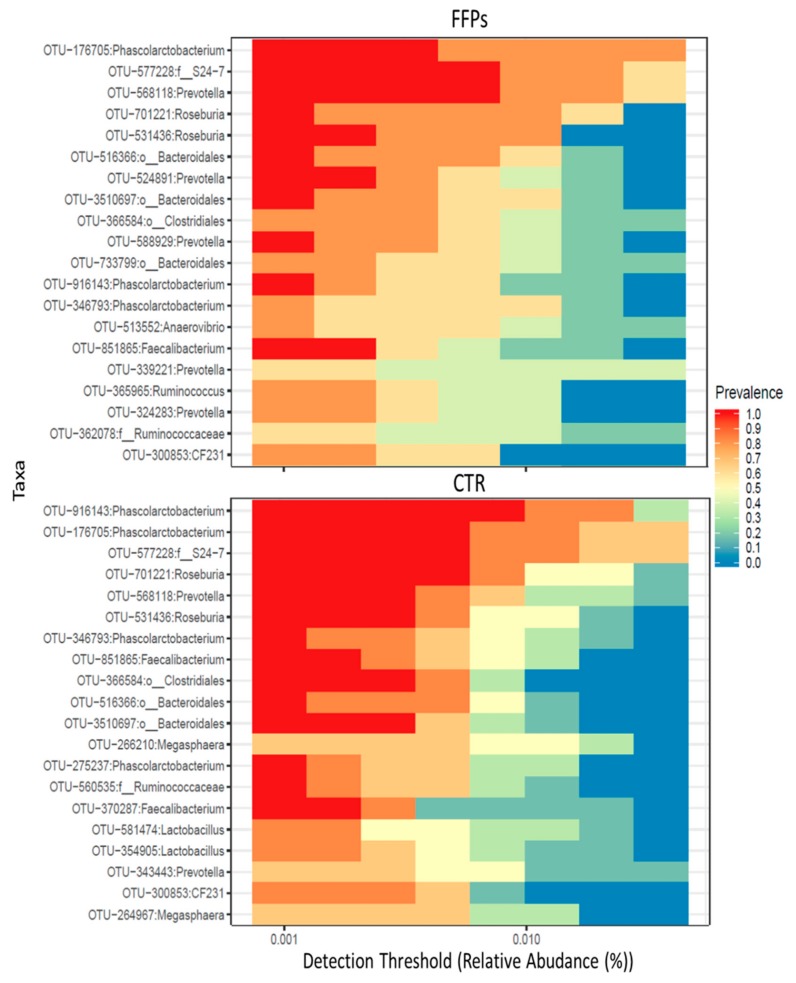
Cross-sectional comparisons of the fecal microbiota in the FFP and CTR groups indicating specific keystone taxa detected at the end of the trial (D 16).

**Figure 7 animals-09-00516-f007:**
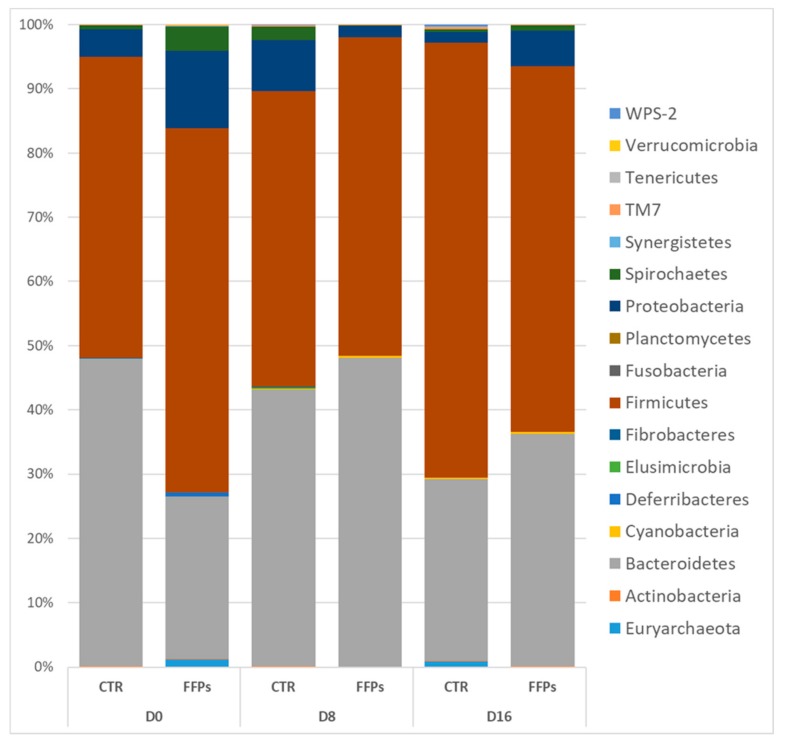
Classification of gene sequences at the phylum level for the control and FFP pigs at each sampling time.

**Figure 8 animals-09-00516-f008:**
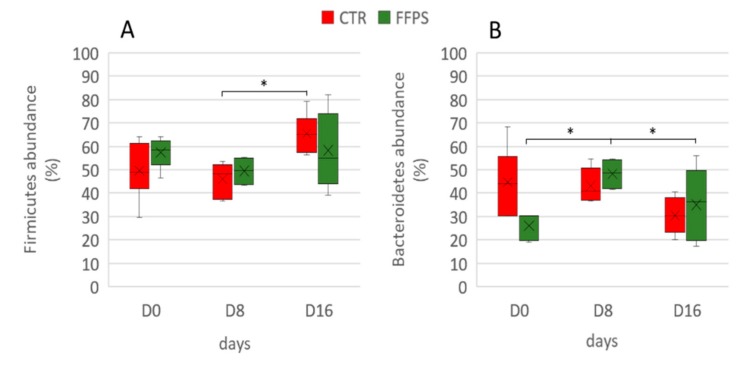
Temporal changes in Firmicutes and Bacteroidetes phyla in CTR (red boxes) and FFP (green boxes) groups. The proportions of Firmicutes (**A**) and Bacteroidetes (**B**) are shown for post-weaning piglets at days 0, day 8, and day 16 of the experiment. The lower boundary of the box is the 25th percentile, the line and the cross within the box mark the mean indicators, and the upper boundary of the box indicates the 75th percentile. Whiskers (error bars) above and below the box indicate the 90th and 10th percentiles; the symbol * indicates *p*-values < 0.01.

**Table 1 animals-09-00516-t001:** Summary of next generation sequencing data and effect of FFP-based diet on diversity and abundance indexes at each sampling time (± standard error of the mean) in post-weaning piglets.

Alpha Diversity Indexes	D 0		D 8		D 16		*p*-Values ^1^		
	CTR	FFPs	CTR	FFPs	CTR	FFPs	T	G	TxG
Shannon’s	6.19 ± 0.19 ^AB^	6.39 ± 0.16 ^AB^	6.46 ± 0.04 ^A^	5.96 ± 0.2 ^B^	6.25 ± 0.10 ^A^	5.76 ± 0.13 ^B^	0.57	0.001	0.71
Chao1	549.71 ± 20.1	556.7 ± 37.8	603.9 ± 16.3	581.2 ± 22.3	609.3 ± 10.8	541.3 ± 22.7	0.67	0.10	0.55
OTUs	470.6 ± 19.4 ^ab^	489 ± 35.2 ^ab^	529.3 ± 14.2 ^ab^	500.0 ± 25.5 ^ab^	534.1 ± 12.1 ^a^	435.1 ± 19.1 ^b^	0.78	0.03	0.16
PD-whole tree	35.3 ± 1.06	36.5 ± 1.67	38.1 ± 0.93	36.4 ± 1.07	37.8 ± 0.72	33.2 ± 1.06	0.70	0.10	0.28

CTR = Control diet, formulated to meet NRC requirements for post-weaning piglets; FFPs = Experimental diet in which 30% of FFPs partially replaced conventional cereal grains. D 0, D 8, D 16 = day 0, day 8, day 16 of sampling, respectively.^1^ Probability values for the effects of Time (T), Group (G) and T X G. ^a,b^ Values within a row with different superscripts differ significantly at *p* < 0.05. ^A,B^ Values within a row with different superscripts differ significantly at *p* < 0.01.

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
