# Peer review of "Influence of Traditional vs Alternative Dietary Carbohydrates Sources on the Large Intestinal Microbiota in Post-Weaning Piglets"

_animals, 2019, doi:10.3390/ani9080516_

Round 1

Reviewer 1 Report

The paper deals with an important topic and provides new information with regards to the use of alternative dietary carbohydrate sources on piglet microbiota, which may be increasingly used in the future. In general, it is a clear paper, and discussion is adjusted to the results. It provides relevant information and novel insights on the topic.

However, there are some aspects which remain unclear or could be improved.

SPECIFIC COMMENTS.

1.       There are a few English/spelling mistakes. Such as:

Line 20.  …as an alternative energy sources (either singular or plural)

Line 72 Formulated instead of formultad

Lines 122-123. “In this study have been used a total of 12 pigs”. Sentence sounds strange. Better “A total of 12 pigs have been used in this study”

Line 165. The is duplicated

2.       Methodological aspects

Line 122-123. 12 piglets were used. It is a low number, understandable according to 3Rs policy and to difficulties in microbiota sampling. However, could the authors provide evidences that the number would be sufficient to obtain statistical significance?

The piglets were housed in individual pens, probably because it was the only way to be able to record individual feed intake. Could the authors provide information on the refinement methods to enhance welfare in those individual pens? (environmental enrichment, possibility of social contact…)

Line 132-135. Could the authors provide more information on how body weight and feed consumption were recorded? Although another paper has been published presenting those results, it is not mentioned until the results section.

Statistical methods. The statistic tests used to analyse performance parameters have not been included (for body weight, ADFI…). Was it a repeated measures test, using individual as experimental unit?

Author Response

We are grateful to the reviewer for her/his comments and suggestions

SPECIFIC COMMENTS.

1.       There are a few English/spelling mistakes. Such as:

Line 20.  …as an alternative energy sources (either singular or plural)

Line 72 Formulated instead of formultad

Lines 122-123. “In this study have been used a total of 12 pigs”. Sentence sounds strange. Better “A total of 12 pigs have been used in this study”

Line 165. The is duplicated

 AU: Mistakes have been corrected. Thank you to pointed them out

2.       Methodological aspects

Line 122-123. 12 piglets were used. It is a low number, understandable according to 3Rs policy and to difficulties in microbiota sampling. However, could the authors provide evidences that the number would be sufficient to obtain statistical significance?

The piglets were housed in individual pens, probably because it was the only way to be able to record individual feed intake. Could the authors provide information on the refinement methods to enhance welfare in those individual pens? (environmental enrichment, possibility of social contact…)

AU: We agree with the reviewer comment. According to RRR approach, number of animal was the lowest possible according to the G*Power  software. In this respect it was also essential to stock animals individually (not in pen) that increased the statistical power using less animals. Taking into account that animal housed individually might have some welfare drawback, all piglets used were able to see/interact (by shared societal view) their conspecific in order to have stimuli an early socialization. Both environmental enrichment and pig-human interactions were also guarantee. Finally, conscious that the number of animal used was limited, the authors decide to state in abstract and conclusion and when needed that "these results should be interpreted with caution since they are case sensitive, ....". All these information have been added to the manuscript. 

Line 132-135. Could the authors provide more information on how body weight and feed consumption were recorded? Although another paper has been published presenting those results, it is not mentioned until the results section.

 AU: Requested information have been added to the manuscript See line 128-132

Statistical methods. The statistic tests used to analyse performance parameters have not been included (for body weight, ADFI…). Was it a repeated measures test, using individual as experimental unit?

AU: Requested information have been added to the manuscript see line 166-170

Reviewer 2 Report

Reviewer(s)' Comments to Author:

Recommendation: Accept after minor revision 

Comments:
In general relevance of the manuscript is high, the use of food industry waste in animal feed is an important aspect for the livestock sustainability. Moreover, the study of the microbiota is nowadays subject of discussion in various fields.
More in details, the aim of the experiment was to investigate the effect of former food as cereal substitute  in pig diets and address its effect on gut microbiota. Quite innovative the subject, i,e, former foodstuffs and their impact on large intestine microbiota investigated by NSG. In general, the study is well designed and results well presented. Nicely the authors do not over stand their results on performance (actually missing in the abstract) but address properly the microbiota aspects.   

Some specific comments 

L 11-26 any comments on performance 

L27-41 as before 

L38-40  rephrase 

On lines 56-57 the sentence is not clear. Please revise it.

On lines 63-64: Other authors evaluated the nutritional composition

[This part L57-63 need e revision since unclear]

L69 during food production 

L74 FFPs (s is missing) 

Line 72: the diets formulated.

Line 85: ingredients

Line 96: such as piglets

Line 118: “in vivo” in italics.

L142 method??

Line 164: remove “The”

I do not understand the sentence at L216 “count per sample abundance of sequences”.

Line 192: “Feed conversion ratio”, remove the capital letter

Discussion comments: The discussion of the results is appropriate and clear. However, the difference between traditional and alternative dietary carbohydrates sources should be better highlighted, in order to obtain a clearer connection to the title of the manuscript.

Specifically, considering the difference between TDF and SDF content in the diets, what is the role of the different content of  TDF, SDF and IDF of the two diets on large intestinal microbiota of the piglets?

L298-304: clarify the sentence.

L292,293,294: in vivo and in vitro in italics.

L 300: DF is the SDF or the IDF?

L344: “whit no effect on colonic bacterial population”

L. 349: has been addressed

Conclusions comments: The conclusions are appropriate to the content of the manuscript.

L 370 decrease or affected 

Author Response

We are grateful to the reviewer for her/his comments and suggestions

AU: Growth performance have been added in the abstract  

L 11-26 any comments on performance 

L27-41 as before 

L38-40  rephrase 

On lines 56-57 the sentence is not clear. Please revise it.

On lines 63-64: Other authors evaluated the nutritional composition

[This part L57-63 need e revision since unclear]

AU: Thank you for pointing out the mistakes. This part has been rephrased.

L69 during food production 

L74 FFPs (s is missing) 

Line 72: the diets formulated.

Line 85: ingredients

Line 96: such as piglets

Line 118: “in vivo” in italics.

L142 method??

Line 164: remove “The”

I do not understand the sentence at L216 “count per sample abundance of sequences”.

Line 192: “Feed conversion ratio”, remove the capital letter

AU: Thank you for pointing out the mistakes. They have been corrected.

Discussion comments: The discussion of the results is appropriate and clear. However, the difference between traditional and alternative dietary carbohydrates sources should be better highlighted, in order to obtain a clearer connection to the title of the manuscript.

Specifically, considering the difference between TDF and SDF content in the diets, what is the role of the different content of TDF, SDF and IDF of the two diets on large intestinal microbiota of the piglets?

AU: Required information have been added to the manuscript. See line 306-311

L298-304: clarify the sentence.

L292,293,294: in vivo and in vitro in italics.

L 300: DF is the SDF or the IDF?

L344: “whit no effect on colonic bacterial population”

L. 349: has been addressed

Conclusions comments: The conclusions are appropriate to the content of the manuscript.
L 370 decrease or affected 

AU: Mistakes have been corrected. Thank you.
